# Sex, Energy, Well-Being and Low Testosterone: An Exploratory Survey of U.S. Men’s Experiences on Prescription Testosterone

**DOI:** 10.3390/ijerph16183261

**Published:** 2019-09-05

**Authors:** Alex A. Straftis, Peter B. Gray

**Affiliations:** Department of Anthropology, University of Nevada, 4505 S. Maryland Pkwy, Box 455003, Las Vegas, NV 89154-5003, USA

**Keywords:** prescription testosterone therapy, hypogonadism, low testosterone, libido, energy, survey

## Abstract

Prescription testosterone sales in the United States have skyrocketed in the last two decades due to an aging population, direct-to-consumer advertising, and prescriber views of the benefits and risks to testosterone, among other factors. However, few studies have attempted to directly examine patient experiences on prescription testosterone therapy. The present exploratory study involved an online self-report survey of U.S. testosterone patients who were at least 21 years of age. The primary focus was on patient perspectives concerning motivations leading to the initiation of testosterone therapy and the perceived effects of treatment. Responses to open-ended questions drew upon a coding scheme incorporating both inductive and deductive approaches, influenced by the clinical, male life history theory, and behavioral endocrinology literature. Results indicated that the most frequent reasons men gave for taking prescription testosterone were low testosterone (37.1%), well-being (35.2%), energy (28.7%), libido (21.9%), and social energy (19.4%); older men claimed libido as a motivation for testosterone initiation more frequently than younger men (*p* < 0.001). Men most frequently claimed testosterone improved their energy (52.3%), libido (41.9%), and muscle (28.5%). Results are interpreted in the context of medical, life history theoretical and behavioral endocrinology approaches, including an emphasis on sex and energy.

## 1. Introduction

The amounts spent on prescription testosterone (PT) and number of prescriptions administered have skyrocketed in recent decades [1,2]. Annual prescription testosterone sales in the United States (U.S.) have increased from $18 million in 1988 to $70 million in 2000 to more than $2 billion in 2013 [3]. An estimated $1.8 billion in 2011 was spent on testosterone sales across some 41 countries, up from $150 million in 2000 [4]. Most of these prescriptions are intended for aging men with low testosterone [1]. The global prevalence for androgen deficiency varies between 10 and 40% and can reach as high as 80% in patients with multiple comorbidities [5]. Treatment with PT has increased worldwide within the last two decades, likely influenced by an aging population. As of 2016, the U.S. represented more than 80% of global PT prescription sales [1,6]. While the rate of new PT prescriptions has dropped in the past 5 years, the U.S. continues to lead in the global PT market share [5,6,7].

Several major factors align with this growth in prescription testosterone. The availability of formulations such as testosterone gels for treatment has expanded. The marketing of testosterone for various conditions packaged under male aging has intensified. Direct-to-consumer advertising via social media and online has enabled bypassing more traditional avenues for clinician–patient discussions about medication and health conditions such as low testosterone [5,8,9]. In addition, unbranded testosterone campaigns and industry-sponsored continuing medical education promotion framed testosterone testing as part of a regular checkup, contrary to the Endocrine Society’s recommendations against general population testosterone screening [10,11,12]. This combination of advertising tactics may have impacted perceptions of what PT can offer an individual with low testosterone, and do not always match data from clinical trials.

The medical literature has sought to address the benefits and risks of exogenous testosterone. PT is given for several indications, often characterized by androgen deficiency and low T blood levels [13,14]. Diagnostic criteria, symptoms, and treatment efficacy can very between forms of hypogonadism, and detection of testosterone levels below a reference range (without a formal diagnosis of hypogonadism). Classical hypogonadism (primary and/or secondary hypogonadism) refers to androgen deficiency due to identifiable congenital (e.g., Klinefelter or Kallmann’s syndrome) or acquired (e.g., trauma, infections, or hyperprolactinemia) disorders in the hypothalamic–pituitary–gonadal axis [13,14]. Symptoms of low testosterone in individuals with classical hypogonadism include failure of secondary sexual characteristics, delayed sexual development, gynecomastia, erectile dysfunction (ED) and others [13,15]. Reduced energy, impaired cognitive ability, increased fat, loss of muscle and disrupted sleep may also accompany low testosterone [13]. Multiple guidelines support providing PT to men with classical hypogonadism to induce and maintain secondary sexual characteristics and correct symptoms of low testosterone [13,16,17]; treatment often seeks to maintain serum testosterone levels at a mid-normal reference range for healthy young adults [14,16]. PT is not recommended for individuals with known prostate or breast cancer, high prostate specific antigen levels, severe lower urinary tract symptoms, high hematocrit, poorly controlled heart failure, severe sleep apnea, and/or on individuals which desire fertility [13,14,16,17].

Much of the medical and societal debate about prescription testosterone refers to its use to treat Low testosterone in aging men. A diagnosis of late onset hypogonadism (LOH) is characterized by consistently low testosterone levels and a combination of symptoms related to androgen deficiency, without an identifiable mechanism other than those related to age or lifestyle [13,16,17]. Wu et al. (2010) found decreased sexual thoughts, erectile dysfunction, and weakened morning erections as the strongest predictors for LOH in a population-based study of 3360 men over 40 [18]. Often, LOH presents with nonspecific symptoms such as: hot flashes, reduced energy levels, decreased bulk, infertility, and impaired cognitive function [19,20]. Many symptoms of LOH may be related to normal features of aging or other chronic illnesses, making PT for individuals with LOH controversial. In 2015, the FDA issued a statement directing testosterone manufacturers to label their products as only approved for use in men with known causes for hypogonadism, after fears of a potential link between PT and cardiovascular risk [20,21]. Contrary to this, multiple society guidelines argue PT is indicated for men with LOH but should be considered on an individual basis, after multiple confirmations of low testosterone and the presence of hypogonadal symptoms [13,16,17,22].

There is a gap in understanding the motives and experiences of men taking prescription testosterone. While insight into clinical benefits and risks and marketing strategies have expanded, we know relatively little about what patients themselves specify about their motivations leading to initiation of prescription testosterone. Comparisons made by Layton et al. (2014) show U.S. men may have different concerns and motivations (such as less concern for sexual dysfunction but more concern over lack of energy) regarding PT compared to patients in the United Kingdom [23]. From a sample of nine interviewees in Canada, Mascarenhas et al. (2016) identified both provider (e.g., whether testosterone was viewed as a legitimate therapy) and patient (e.g., access to information about effects of testosterone) insights into why men might seek prescription testosterone [24]. Here, we employ an exploratory online survey featuring open-ended questions answered by a convenience sample of U.S. men 21 years of age and older on prescription testosterone. We do not purport to provide representative, generalizable data for all U.S. men on prescription testosterone, though we do broadly address whether men’s self-reported experiences on prescription testosterone are consistent with prescribing patterns in the literature. Our aim is to address the central questions: Why do men initiate prescription testosterone therapy? What do they perceive to be the benefits and side effects of prescription testosterone? We also seek to contextualize patient experiences in learning about testosterone and how they manage their therapy through solicitation of additional open-ended and close-ended items. Lastly, we ask whether the reasons and experiences for taking prescription testosterone differ by male patient age. Drawing broadly on the current literature, we expected libido, energy, muscle, mood, and fat loss to be the most frequently given reasons why men take prescription testosterone and the benefits they perceive from it. Aging biases life history strategies towards maintenance and repair functions (such as maintaining health) over expensive competing allocations such as anabolism and mate-acquisition. Thus, as men age, they may be less motivated to acquire mates and compete with conspecifics (through strategies such as gaining muscle), unless their immediate physiological and affiliative needs are met. Drawing on life history theory (also see: the Ivanov et al. (2018) content analysis of PT websites), it was hypothesized that ages of men would also influence motivations and benefits for taking prescription testosterone, with older men emphasizing regaining youth, health, and wellbeing as reasons for PT initiation and younger men more concerned with improving sex, competition, and muscle [12,25,26,27].

## 2. Methods

### 2.1. Study Design and Data Collection

We recruited a convenience sample of male respondents 21 years of age and older by sampling from social media and PT forums’ email lists. PT groups were located using Facebook’s search function and the search terms “PT, testosterone therapy, low T, hypogonadism, and hormone deficiencies”. Recruitment posts and emails were distributed among groups that focused on prescription testosterone, shared peer-reviewed papers on testosterone therapy, specialized in LOH or a form of classical hypogonadism and/or hormone deficiencies. Groups focusing on bodybuilding and/or performance enhancement through nonprescription methods were excluded. During the consent process, participants were asked to confirm their age, sex, current PT prescription, and country of residence. Inclusion criteria were being male, 21 years old (or older), resident in the U.S., and currently on a form of prescription testosterone. Individuals who received their prescription abroad or identified as transitioning to male were excluded and outside the scope of the present study. Recruitment and data collection began after the study was deemed exempt by the University of Nevada, Las Vegas Biomedical IRB (protocol # 1146420-3). Written consent was given prior (via check box and reCaptcha) to the beginning of the online survey; identifying information (e.g., name) was not recorded.

Data were obtained through use of an online survey (S1 Qualtrics survey instrument). The survey included questions related to demographics (e.g., age, employment, education, family/marriage status), clinical characteristics (e.g., current or prior medical conditions, type of PT formulation, source of PT prescription, and duration on PT), perceived changes after taking PT (e.g., “Has your work motivation changed since taking prescription testosterone?”; if respondents chose “yes” they would be asked to explain how in an open-ended question) and 5-point Likert items to gauge aspects such as relationship satisfaction (e.g., “How satisfied are you with your current relationship/marriage status?”, rated from “extremely satisfied (5)” to “extremely dissatisfied (1)”). Survey questions were inspired and partially adapted from the semi-structured interview guide in Mascarenhas et al. (2016) [24]. In addition, five open-ended questions (“How did you learn about prescription testosterone?”; “Why did you decide to take prescription testosterone?”; “What did you perceive as the benefits of testosterone?”; “Please explain any side effects or concerns you have about your testosterone prescription?”; “Describe when you first began to notice the effects of prescription testosterone”) were included toward the end of the survey to directly address our research questions. All respondents were given the option to leave blank any questions they were uncomfortable answering, resulting in a slightly smaller sample size of respondents for some items.

Several steps were taken to try to ensure data integrity. A reCaptcha was placed at the end of the consent form and at the end of the survey to minimize fraudulent attempts. The “prevent ballot box stuffing” option was selected on the Qualtrics survey-hosting platform to prevent multiple attempts from similar IP addresses (IP addresses were anonymized by Qualtrics). Further attempts to prevent survey fraud were undertaken based on a 2015 review on fraud in Internet research [28]. For example, participants navigated multiple pages to complete the survey, and time spent on each page was recorded. Survey responses were monitored daily to identify fraudulent patterns in survey responses (such as identical responses re-used in multiple surveys).

### 2.2. Measures

Survey responses were self-report, retrospective, and anonymous. Participants’ ages were recorded as mean-centered. Additional questions were categorical and utilized combinations of multiple-choice close-ended questions, or open-ended text entry. Five questions were 5-point scaled items (e.g., “Do you hope to have children (biological or adopted) in the future?”) and were used to discern the extent a participant felt about specific items. Open-ended questions were manually coded using inductive first-cycle coding methods [29]. The initial codebook was adapted from Ivanov et al. (2018) [12]. Codes represented a combination of inductive and deductive concepts, with deductive concepts largely based on the clinical testosterone literature referenced in the introduction and codes such as social energy, mate seeking, relationship and dominance from evolutionary life history literature. For example, low testosterone was coded any time a participant explicitly mentioned low testosterone levels as a factor in their responses. This could have been an explicit blood value (e.g., <280 ng/dL) or a subjective belief that a participant’s testosterone level was lower than optimal. As this study did not explicitly ask for total testosterone blood values, and the criteria for androgen deficiency vary between labs and medical societies, we coded participants’ beliefs of low testosterone rather than developing an objective cut-off. All coding was performed by the first author. Further information on codes used for this study can be found in the codebook provided in Table 1.

To address our research questions, we recorded the frequencies of themes found in the self-report questions “How did you learn about prescription testosterone?” and “Why did you decide to take prescription testosterone?”, as well as patterns found in close-ended responses concerning diagnosis and current/prior health conditions. The responses were then analyzed based on age group (≤39, *n* = 52; ≥40 *n* = 53) to test for age-related differences in the above-mentioned frequencies. The remaining open-ended questions asked men what effects they noticed after taking PT (general benefits, side effects, effects on work motivation, effects on family life and onset of first effects). Once again, frequencies were compared between younger and older men.

### 2.3. Statistical Analyses

Due to the exploratory nature of our survey, no prior power testing was conducted. Many values compared in this study are categorical in nature. Fisher’s exact test was used to calculate *p* values in categorical data to compare differences between <40 and ≥40 individuals, with α = 0.05. Mann–Whitney U tests were employed to test for differences in Likert-based responses between younger and older men.

## 3. Results

### 3.1. Sample Characteristics

Table 2 presents demographic characteristics of survey respondents separated by age (<40 and ≥40) and overall. The final sample included 105 participants (µ = 32.22, SD = 4.46). Half (49.5%) of the sample was under the age of 40, and the remainder above. Most participants were Caucasian (82.9%), employed (90.5%), married or in a committed relationship (78%) and had children (68.6%). In total, 89.5% of respondents reported they were somewhat to very sexually active, and 80.1% rated their desire to be more sexually active above neutral (27.6% chose “probably yes” and 53.3% chose “definitely yes”, when asked to rate “do you want to be more sexually active?”, on a 5-point scale); 96.2% of older men scored their desire (to be more sexually active) above neutral vs. 65.3% of younger men (*p* ≤ 0.001).

### 3.2. Clinical Characteristics

Table 3 summarizes the clinical characteristics of participants, separated by age group and totaled. The most frequently coded reasons for testosterone being prescribed did not list hypogonadism as a diagnosis. Patients described, in open-ended responses, that they received PT from a physician for the following reasons, excluding a hypogonadism diagnosis (coded as “other”): low testosterone (48.6%), energy (10.5%), and libido (17.8%). Injection formulations were the most common form of PT in the group (81.9%). Additionally, 11.5% of younger men took an oral formulation (compared with 0 older men *p* = 0.013). A majority (79%) of the sample reported either a current or prior chronic condition, other than androgen deficiency. Fifty-four percent reported two or more comorbidities, and older men represented 77.3% of these responses (*p* = 0.001). The most frequently reported conditions were: high blood pressure (25.9%, 15% younger vs. 37.7% older, *p* = 0.013), sleep apnea (21.9%, 11.5% vs. 32%, *p* = 0.017), obesity (20%, 11.5% vs. 28.3%, *p* = 0.05), depression (18%, 17.3% vs. 18.8%, *p* = 1), and high cholesterol (17.1%, 5.7% vs. 28.3%, *p* = 0.003). Hypothyroidism (3.8% vs. 16%, *p* = 0.052) and hypertension (15.2%, 7.6% vs. 22.6%, *p* = 0.055) exhibited trends toward higher frequency among older men.

### 3.3. Sources of PT and PT Information

A total of 148 codes emerged when participants (*n =* 105) responded to “How did you learn about prescription testosterone?” Participants reported that they most frequently learned about testosterone from a licensed health care provider (*n* = 54, 51.4%). Participants also frequently reported online sources (*n* = 34, 32.3%) and word of mouth (*n* = 20, 19%) as sources of their testosterone information. Additionally, participants specifically mentioned social media (*n* = 19, 18%) and offline direct-to-consumer advertising (*n* = 13, 12.3%), such as television commercials, as information sources. Results are in line with our expectations that the Internet and licensed health care providers (often primary care givers) are primary sources for prescription testosterone information.

From 105 respondents (52 younger men and 53 older) who answered the question “Why did you decide to take prescription testosterone?”, 249 coded responses emerged. Table 4 presents the frequencies of codes found during our analysis. The most frequently recorded codes were related to general mentions of low testosterone (*n* = 39, 37.1%) and well-being (*n* = 37, 35.2%). Codes related to improving energy or reducing fatigue were also frequent (*n* = 30, 28.7%). Respondents also described libido (*n* = 23, 21.9%) and social energy (*n* = 20, 19.4%) at high frequencies. Results partially supported our expectations about key themes guiding men’s motivations to initiate prescription testosterone therapy.

Age-related trends were contrary to our expectations. Libido codes were more than four times as frequent among older men (7.7% vs. 35.8%, *p* = 0.001), as opposed to our prediction that younger men would recognize more sex-related codes. Other/miscellaneous reasons may trend more with younger men (19.2% vs. 7.5%, *p* = 0.092). Differences in dominance (3.8% vs. 0%, *p* = 0.243), erectile dysfunction (1.9% vs. 7.5%, *p* = 0.363), muscle (3.8% vs. 7.5%, *p* = 0.678), well-being (38.5% vs. 32.1%, *p* = 0.544), and youth (1.9% vs. 3.8%, *p* = 1) could not rule out the null hypothesis. Larger sample sizes may be able to detect differences between our less-frequently coded themes.

### 3.4. Effects of Testosterone Observed by U.S. Men

Of 105 respondents, 68.5% reported that they noticed the effects of PT within the first 3 months of treatment, and most frequently reported that they felt its effects between 2 and 4 weeks (*n =* 35, 33.3%). Age-related differences were identified in men who did not recall or notice when effects of PT occurred (*n* = 21, 20%). Younger men could not recall or did not notice the effects of PT more than three times as frequently (*n* = 16, 30.8% vs. *n* = 5, 9.4%, *p* = 0.007). Table 5 presents the frequency of codes recorded from responses (*n* = 319 coded from *n* = 105 respondents) to the open-ended question “What did you perceive as the benefits of testosterone?”.

Most respondents claimed they perceived benefits after taking PT (*n* = 64, 61.0%), whereas approximately 39% did not claim any benefits; this did not vary by age (*p* = 1). Among the overall group, the most frequently coded benefits after taking PT were: energy (52.4%), libido (41.9%), and muscle (28.6%). Other/Misc. benefits (25.7%), well-being (18.1%), fat (18.1%), and social energy (17.1%) were coded relatively often. Age-related differences for this question were all found to have *p*-values > 0.10.

Respondents were also asked to describe changes in either their work motivation or family life after taking PT. Of those who responded (*n* = 97) to “Since taking prescription testosterone, have you noticed any changes in your motivation at school/work?” *n* = 78 (80.41%) claimed they noticed effects, and 98% claimed those effects were positive. The most frequently coded themes among those who reported changes in motivation (*n* = 78, 74.3%) were: social energy (*n =* 32, 30.5%), energy (*n* = 31, 29.5%) and focus (*n* = 17, 16.2%). Only social energy showed substantial age-related differences (*n =* 8, 7.6% vs. *n* = 24, 22.9%, *p* = 0.004). When presented with the item “Please describe any changes you have noticed in your family life, since taking testosterone”, *n* = 105 respondents answered and *n =* 61 (58.1%) reported positive effects due to PT (the remainder either responded with negative effects or “n/a”). Social energy (*n* = 25, 23.8%), energy (*n =* 20, 19%), and libido (*n* = 20, 19%) were most frequently identified. No strong age-related differences were found between older and younger men in the changes they experienced in their family lives.

Lastly, participants were asked to explain any side effects or concerns they may have about their testosterone prescription. Of *n* = 105 responses, 39% had no concerns or side effects to report. Elevated estrogen (*n* = 13, 12.4%) was the most reported concern/side effect among these men. Other notable side effects/concerns were acne (*n* = 12, 11.4%), hypothalamic–pituitary–gonadal axis disruption (*n =* 12, 11.4%) and elevated hematocrit/blood cell count (*n* = 10, 9.5%). Strong age-related differences were not found among reported side effects or concerns.

## 4. Discussion

To our knowledge, this study was the first attempt to directly ask U.S. men, in targeted open-ended questions, why they initiated PT and what effects they noticed from treatment. Overall, men most often reported they sought PT to improve low testosterone, general well-being, energy, and libido; however, improving libido was specified more often among older men. Additionally, men frequently reported positive effects after taking PT, both in general and when asked to contextualize these effects within their work motivation and family life; effects frequently stated were improvements in energy, libido, and muscle. Most participants reported no negative side-effects or concerns after taking PT, and most participants would recommend PT to friends and family members. Further, findings mirror the general trends and prescribing patterns found in the U.S. PT literature [2,7,13,23,30,31,32,33,34].

The emphasis on low testosterone as the most common reason why men in the sample sought testosterone implies a problem in need of a solution. It follows that PT is perceived as the direct solution to low testosterone by individuals who have or believe they have low testosterone. Several factors may lead to low testosterone in men. Multiple guidelines agree on PT for men with classical hypogonadism to induce and maintain secondary sexual characteristics and correct symptoms of androgen deficiency [13,16,17,35]. Alternatively, the diagnosis of LOH (characterized by consistently low testosterone levels and combinations of symptoms related to androgen deficiency but without identifiable mechanisms other than those related to aging or lifestyle) [13,16,17] is prevalent among western men [1,23]. While diagnostic criteria for classical hypogonadism seem clear [13,36], diagnosis of LOH varies by specialist, lab, medical society, and region [13,16,17,24,35,37]. Furthermore, much of the expansion of U.S. PT sales has been driven by off-label indications such as LOH or treatment of androgen levels without a reported diagnosis [7,31].

In the present study, 80% of respondents reported that they were prescribed testosterone for an off-label indication (LOH or “other”), making interpretation of low testosterone problematic and only five cases (1.06%) explicitly mentioned their hypogonadism diagnosis as a reason for initiating PT. Men under 40 more frequently reported “other” as the reason they were prescribed PT (*p* = 0.01) instead of other diagnoses; this may indicate that prescribers (most often family doctors/general practitioners in this sample) are more hesitant to diagnose an individual under 40 who presents with similar symptoms and blood values as those with LOH. While a full review of the androgen deficiency literature is beyond the scope of this paper, it is notable that multiple publications note inconsistent prescriber pretreatment and follow-up evaluations (see [1,23,32]), which may lead to PT prescriptions for a wide range of testosterone levels. Further complicating interpretations of the low testosterone self-report responses are the variable effects PT (e.g., ranging from overall improvement to no apparent effects) may have between eugonadal and hypogonadal men [13,38,39,40,41,42,43,44,45,46]. Future patient surveys may attempt to document total and free testosterone levels (and other self-report recollections about low testosterone testing) to better contextualize individual factors (i.e., low testosterone vs. low- normal testosterone in an older male) that may add to the interpretation of men’s self-report claims.

Since improving general well-being was the second most reported theme, it can be interpreted that men want PT to solve their problem of low testosterone and generally improve their overall lives. Improvements in quality of life have been reported in patients receiving PT for hypogonadism and LOH [47,48] but these effects can vary based on study design [49]. Furthermore, quality of life and well-being in PT patients is likely composed of several context-dependent factors (e.g., improvements in sexual functioning would likely improve the quality of life in a sexually active and married individual but not someone who was sexually inactive before they had low testosterone) and is relative (i.e., determined by the decline in quality of other untreated-low testosterone patients and/or healthy individuals in a respondent’s social network). Higher frequencies of more specific (e.g., energy (*n =* 55) or libido (*n* = 44) effects noticed after PT in this survey suggest that there may have been an underlying reason when men responded with well-being, one which may be dependent on life-context, but was missed due to the survey design. Future efforts should probe for more contextual information regarding low testosterone and well-being to further elucidate the underlying relationships between these two themes.

Improving energy was the third (28.6%) most frequently coded reason for why men sought testosterone and the most frequently reported benefit they associated with treatment (52.4%). Additionally, energy (social energy and energy) related themes were also most common when individuals were asked to describe changes in their work motivation and family lives. Analysis of our open-ended responses revealed that energy could be referring to as somatic (e.g., energy which combats fatigue and permits physical activity), social energy (e.g., motivation or drive to engage in a social activity such as competition in the workplace or a general ‘zest’ for life) or a combination of both. We identified energy as social energy in 20 cases of why men sought PT, and this became more common in a social context such as at work or in family life. This suggests that the benefits men seek to achieve from testosterone (and the effects they ultimately notice from it) might be modulated by individual circumstance such as physical demands and/or high levels of social competition required by a career. Distinguishing between these two types of energy and the context when they may be desired is essential to understanding why men in this sample took PT and the primary benefits they claimed to notice from it.

The causal effects of exogenous testosterone on energy are not clear. Lasaite et al. (2017) found no improvements in emotional state or quality of life after 2 years of treatment in a small sample of young and middle-aged hypogonadal men [50]. Neither the testosterone trials nor a commissioned systematic review and meta-analysis found improvements in energy (measured as fatigue) versus placebo [13,41]. Other studies have found a lack of significant benefits on factors such as depression, cognition, energy, and mood [51,52,53]. Testosterone levels may attenuate age-related aerobic decline, which may be felt as energy in an aerobically active individual, but these relationships have not been investigated. Several other factors such as multimorbidity (an attribute frequently observed in our ≥40 group) may contribute to weakness in young and old men independently of testosterone levels [54,55], confounding our interpretation. In sum, the clinical and experimental literature on testosterone’s role in regulating men’s energy levels is mixed. This seems to be contradicted by the self-report responses of more than half the men in our survey. A large body of literature studies testosterone’s role in modulating social behavior in males [56,57,58,59,60,61,62,63,64,65]. While much of this empirical work is correlational in research design [58], recent studies have evaluated the effects of one-time exogenous testosterone administration and found a potentially causal effect on the influence of testosterone levels (in males) on behaviors related to: aggression [62,63], cooperation [64], and perceptions of their own physical dominance [65].

While this literature is too extensive to review here, it highlights the role of testosterone in the immediate social context of a male. The high prevalence of energy and social energy can be interpreted as men’s’ desire to seek and perceive a more active engagement in their world, including areas such as their family and competitive employment environments, although randomized double-blind trails would need to confirm this in a more robust manner. Most of this sample was employed, in a committed relationship, and had dependents; this indirectly suggests a group of individuals who are socially engaged to a substantial degree. In an empirical contribution to this conversation, Rosen et al. (2018) surveyed hypogonadal patients to determine characteristics which led them to initiate PT and found “deterioration of work performance” as a statistically significant factor, suggesting that contextual factors (e.g., work performance) may be more significant than measures of “energy” [33]. Overall, our responses indicate a nuanced relationship between testosterone and energy. Future efforts can attempt to probe for the context in which a participant desires energy (e.g., energy to remain physical active in a gym vs. energy to go on a date with a spouse). Additionally, random-double blind trials could attempt to consider the effects family might have on PT’s effects, although discussion of this is out of the scope of this paper.

Libido was frequently reported as a factor related to PT initiation and was the second most reported benefit after treatment; additionally, it was frequently reported as an improvement in men’s family lives. Interestingly, men over 40 reported desire to improve libido as a reason they sought PT more than three times as frequently as younger men. This is contrary to information found on testosterone manufacture websites, which primarily focus advertising related to sex to men under the age of 40 [12]. An interpretation is that the sample of participants in the present study was comprised primarily of men between the ages of 28 and 48 years, indicating that most of these men are either young adults or in their early-middle age. Most of these men are also married (and more of the older group are married), and the majority of sexual activity in humans occurs in committed pair bonds [66]. These slightly older men may be facing an inability to maintain a level of desire they feel is required by their sexual partner (or the subjective desire they feel they should have at this stage of their life), who, if in a similar age range, may have only just begun facing similar declines in libido [67,68]. Overall, these findings mirror the aging male literature which finds that older men with low T are most accurately diagnosed with the presence of three symptoms of sexual dysfunction (libido being one of them) and low testosterone blood levels [13,18,19]. Future studies can gauge self-report libido in PT patients aged over 60, who already may face altered sexual desire [69].

Both young and older PT patients frequently reported improvements to libido both in general and when specifically asked about how PT affected their family lives; this is consistent with the clinical literature. A systematic review and meta-analysis of randomized placebo-controlled trials showed improvements in libido, erectile function (ED) and sexual satisfaction among men given testosterone compared to placebo [48]. Results from the testosterone trials suggest that PT increases all aspects of sexual activity and libido compared to placebo, although its effect on ED was weaker. Moreover, levels of sexual activity and libido were associated with incremental increases in testosterone blood values [41]. Finally, a 2014 meta-analysis on PT and sexual function concluded that testosterone supplementation has positive effects on sexual function in hypogonadal subjects [70]. Overall, the above studies and reviews suggest that PT can improve sexual activity and libido in men with low testosterone, and this appears consistent with our self-report data. Further information on testosterone blood values and other measures of sexual function can help explain these effects in more details. Future studies can also attempt to gauge perspectives of the sexual partners of PT patients. This can help elaborate the degree to which libido was affected before and after treatment, and how this is matched with and/or is perceived by a patient’s sexual partners.

Other results from this survey mirror recognized patterns in the clinical testosterone literature. Many of the side effects and concerns noted in our survey are commonly reported among PT patients [13,34,71]. High hematocrit, high estrogen, increased acne, and gynecomastia were reported by a small portion of our sample. This may be due to treatment-related information (e.g., dosages) or related to the type of PT formulation. Most of our sample reported being on an injection form of PT formulation. Intramuscular injections of testosterone can come in a variety of forms based on their half-lives [34]. Short-lasting PT formulations are known to cause fluctuations between super and supra physiological levels of testosterone and may account for some of the side effects noticed by this group [34,71]. Future studies can attempt to more accurately gauge PT formulation and treatment compliance and relate it to the effects noticed after treatment. Overall, men did not report significant side effects or concerns after PT. With the addition of a portion of men receiving PT from men’s clinics, the studied group appeared to share similar features with U.S. PT patients in recent years [1,2,23]. Further research could also be advanced to identify variation in PT patient’s motivations in international settings given variation in prescribing patterns, economies, work, social, and family roles.

Limitations of this study are the small convenience sample and self-report retrospective observational data. Causal relationships or clear generalizability cannot be determined from this exploratory survey design and sampling strategy. That said, we found a saturation of themes related to prescription patterns and overall experiences which mirror trends known in the PT literature, suggesting that the findings presented here may be relevant. This sample may also be overestimating both positive and negative experiences related to PT due to the self-selected nature of the survey. Efforts to recruit from a broad sample of online sources were undertaken to minimize sampling bias. That said, this study mostly surveyed white middle-aged adults who were at least middle-class. Future studies can attempt to diversify the participant pool to capture the experiences of PT patients in a larger range of demographics. One such effort could be to record a patient’s sexuality, which may have important interactions with how an individual experience seeking medication such as PT and the effects this treatment may have between individual contexts and sexual histories [66,72,73]. Finally, the survey length may have led to a higher drop-out of individuals. That said, we chose to compensate our participants and this likely helped alleviate this to a degree.

The strengths of this study include the novel research design of gauging prescription testosterone patients’ experiences in a set of core open-ended questions and applying behavioral endocrinology principles to the interpretation of our data. Prior studies have evaluated advertiser and clinical perceptions of PT, but few have assessed men’s experiences directly. Prior surveys such as Rosen et al. (2018) investigated a small sample of hypogonadal patients but found limited results due to the study design [33]. To our knowledge, this survey is the first attempt to ask PT patients to report their experiences without being prompted by closed-ended questions. This revealed contextual information, such as the role of social experience, which would not have appeared in a more quantitative study design. Future efforts can attempt to repeat findings found here among a larger and more diverse sample. Alternatively, efforts can be made to interview patients for more qualitative open-ended responses, which can help clarify some of the nuance in PT’s effects on men and guide future clinical research.

## 5. Conclusions

In summary, our online sample of 105 PT patients found that men most frequently sought PT to address problems related to energy, libido, social energy, and well-being. Effects noticed by PT patients were mostly positive and consisted of increased energy, libido, and muscle. Further investigation of the role of energy revealed a social role of male energy, which may involve the cooperative and competitive interactions of their day-to-day life. Further work should be undertaken to examine how aging may alter the social context (e.g., [66,72,73]) in which PT is experienced and how PT itself may alter how social context is experienced. At the moment, it appears that these PT patients sought and found a source of well-being, energy and sexual interest, with some muscle and less fat too.

## Figures and Tables

**Table 1 ijerph-16-03261-t001:** Coding scheme for open-ended survey responses.

Parameter	Definition	Examples
Anxiety	Explicitly mentioned anxiety. Can reference a diagnosed condition or suspicion of the presence of anxiety. Excludes general complaints of stress or tension.	“I have diagnosed anxiety issues”.
Cancer	A form of cancer which may cause low testosterone—either directly or during treatment.	A diagnosis of testicular cancer.
Classical Hypogonadism	Refers to androgen deficiency due to identifiable congenital (e.g., Klinefelter or Kallmann’s syndrome) or acquired (e.g., trauma, infections, or hyperprolactinemia) disorders in the hypothalamic–pituitary–gonadal axis. A formal diagnosis of hypogonadism. Excludes late onset hypogonadism or no diagnosis.	“Secondary hypogonadism”.
Clinic	A licensed prescriber of testosterone that is not the primary caregiver or a specialist (such as a Urologist). Can include a men’s or hormone clinic.	“I received testosterone from entourage medical, a hormone clinic”.
Cognition	General feelings of improved mental accuracy, energy, clarity, fortitude, problem solving and/or capacity without explicit mention of social context. This excludes focus.	“Mentally sharper and more alert.”
Depression	Explicitly mentioned depression. Can reference a diagnosed condition or suspicion of the presence of depression. Excludes complaints of sadness or moodiness.	“I wanted to help my depression”.
Doctor	Reference to either a primary (or specialist) care giver’s recommendation or prescription of prescription testosterone (PT).	“My doctor prescribed an ...”
Dominance	Assertiveness and or feelings of power and influence over others. References to feeling powerful or strength.	“I’ve become slightly more assertive, where I was previously afraid of “rocking the boat.”
Direct-to-Consumer Advertising	Reference to direct-to-consumer advertising. Excludes online sources of ads.	“Advertisement for a local hormone clinic.”
ED	Inability to maintain an erection sufficient for satisfying sexual activity. Excludes libido.	“Difficulty retaining an erection”.
Energy	General mentions of energy without descriptions of context and/or references to motivation, focus, tolerance, mindset and/or drive in a social context. Physical energy required to complete a task. Physical endurance and/or stamina. The opposite of fatigue.	“More energy means more available to do.”
Family	Information from family. Excludes other word of mouth information (friends).	“…research and I have a nephew in medical school.”
Fat	Descriptions of gaining weight or fat. A desire to lose weight or fat. Excludes lean mass gain.	“…I started to gain enormous amounts of weight.”
Focus	Explicit mention to focus, without mention of social context or other improvements in cognition.	“I am more focused.”
Injection	Explicit mention of an injectable form of prescription testosterone.	“1–3 months into my first injection.”
LHCP	Any licensed health care practitioners, who prescribe TRT.	“…from my primary physician”.
Libido	Sexual desire. Wanting or seeking sexual behavior. Arousal during sexual scenarios. Excludes ED.	“Low sex drive”.
Low Testosterone	General reference to low testosterone blood levels. Can also express a desire to adjust testosterone levels to a more ‘adequate’ level. This may be an objective measure of testosterone blood values with a subjective account of what optimal testosterone levels may consist of. Low testosterone was not determined by the coder without a participant’s belief of the presence of Low T.	“Low testosterone levels and had all symptoms of low testosterone.”
Mate Seeking	A desire to start or seek out a sexual relationship(s). Excludes codes for sex, libido, and ED.	“…and date more people so I decide to take it.”
Mood	General improvements in mood or mindset without reference to context.”	“I am more positive.”
Muscle	Descriptions of muscle or gaining muscle. Complaints about losing muscle. General references to muscle. Excludes references to fat weight.	“I noticed I was making gains.”
Negative Effect	A negative effect of PT not related to an explicitly mentioned medical condition or known side effect.	“I am less motivated at work.”
Online	General reference to online sources of information or advertisements, without specific mention of a social media. Can include search engines.	“Google.”
Other	Information from other sources or unspecified sources. Examples include self, unspecified research, unanswered question.	“I did research”.
Relationship	Any description of taking testosterone due to an ongoing relationship. Excludes libido, ED, and Sex.	“My wife pushed me into the doctor’s office to have the conversation.”
Sex (other)	General comments about sexual behavior/sexual functioning life which do not fit within the Libido, relationship, or ED codes.	“…last longer in bed.”
Social Energy	A combination of factors referring to motivation/confidence/engagement/improvement in a social context. Energy, motivation, mindset, tolerance and/or drive to required participate or engage in a social setting or context (i.e., relationship, work, school, social network). Energy when referring to drive, or motivation in a social context. This excludes physical energy (i.e., energy to get out of bed or finishing a task) or energy in an unspecified context. Excludes solitary mention of energy.	“I had almost no desire to achieve anything at work or personal life. Shortly after starting PT, I started looking at goals we could achieve at work and was able to exceed my expectations and earn a bonus.” “I have more energy to be a good father.” “I have more drive at work.”
Social Media	Reference to online forums, groups, social networks Can include blogs. Excludes online ads, media, or scholarly sources.	“Forum on anabolic pointed me in the direction of seeking blood tests.”
Substance Abuse	Explicit mention of substance abuse, which may dysregulate the HPG axis. Can be prohormones, alcohol, opioids, or other drugs.	“I abused pro-hormone ‘epistane’ in 2012 …”
Well Being	Related to general improvements (or desire to improve) health and/or well-being. Only used when specific symptoms are not mentioned but an emphasis on health is expressed.	“To feel better”.
Word of Mouth	Information from word of mouth. Can include friends or community members. Excludes family, doctors, and social media.	“From people at my gym.”
Youth	To maintain or regain youth. Can also refer to “feeling aged”. This excludes preventing age related mortality or illness.	“19 years ago I was 33 and I just didn’t feel the way I thought a 33 years old healthy man should feel.”

Codes were created inductively and deductively using concepts from the clinical testosterone therapy, behavioral endocrinology, and human life history literature [12,30]. The table shows codes reported in the below results, how these codes were defined, and an example of a statement which generated the code.

**Table 2 ijerph-16-03261-t002:** Demographic data of PT survey respondents.

Parameter, (%)	Respondents <40 Years (*n* = 52)	Respondents ≥40 Years (*n* = 53)	Totals (*n* = 105)
Age			
Range	21–39	40–66	21–66
µ (SD)	32.22 (4.46)	45 (6.23)	39.91 (8.87)
Ethnicity			
African American/Black	4 (7.7)	0 (0)	4 (3.8)
Caucasian	40 (76.9)	47 (88.7)	87 (82.9)
Hispanic/Latino	5 (9.6)	2 (3.8)	7 (6.7)
Other	3 (5.8)	4 (7.5)	7 (6.7))
Household Income			
Less than $25,000	4 (7.7)	3 (5.7)	7 (6.7)
$25,000 to $49,999	9 (17.3)	8 (15.1)	17 (16.2)
$50,000 to $74,999	19 (36.5)	6 (11.3)	25 (23.8)
$75,000 to $99,999	9 (17.3)	12 (22.6)	21 (20)
$100,000 to $199,999	8 (15.4)	18 (33.9)	26 (24.8)
$200,000 or more	2 (3.8)	5 (9.4)	7 (6.7)
Did not disclose	1 (1.9)	1 (1.9)	2 (1.9)
Highest level education			
<High school	0 (0)	1 (1.9)	1 (≤1)
High school degree	11 (21.2)	8 (15.1)	19 (18.1)
Some college	17 (32.7)	18 (33.9)	35 (33.3)
Vocational school	6 (11.5)	4 (7.5)	10 (9.5)
Bachelor’s Degree	15 (28.8)	13 (24.5)	28 (26.7)
Graduate/Professional Degree	3 (5.8)	9 (16.9)	12 (11.4)
Currently employed	45 (86.5)	50 (94.3)	95 (90.5)
Married/Committed	35 (67.3)	47 (88.7)	82 (78.1)
Has children	29 (55.8)	43 (81.1)	72 (68.6)

Columns are expressed as: count (%), unless noted elsewhere. Percentages in rows have been rounded to the nearest 10th.

**Table 3 ijerph-16-03261-t003:** Clinical characteristics of survey respondents.

Parameter, (% of Respondents)	Respondents <40 Years (*n* = 52)	Respondents ≥40 Years (*n* = 53)	Total (*n* = 105)
Hypogonadism diagnosis			
Classical hypogonadism	5 (9.6)	12 (22.6)	17 (16.2)
Late onset hypogonadism	10 (19.23)	18 (33.9)	28 (26.7)
No answer	3 (5.8)	1 (1.9)	4 (3.8)
Other	34 (65.50)	22 (41.5)	56 (53.3)
PT provider			
Endocrinologist	11 (21.2)	15 (28.3)	26 (24.8)
Urologist	8 (15.4)	8 (15.1)	16 (15.2)
Family doctor	20 (38.5)	20 (37.7)	40 (38.1)
Hormone clinic	9 (17.3)	8 (15.1)	17 (16.2)
Telemedicine	1 (1.9)	1 (1.9)	2 (1.9)
Other	3 (5.8)	1 (1.9)	4 (3.8)
PT Formula			
Gel	4 (7.7)	6 (11.3)	10 (9.5)
Injection	40 (76.9)	46 (86.8)	86 (81.9)
Pellets	2 (3.8)	1 (1.9)	3 (2.9)
Oral	6 (11.5)	0 (0)	6 (5.7)
Length of time on PT			
<1 month	2 (3.8)	1 (1.9)	3 (2.9)
1–6 months	18 (34.6)	13 (24.5)	31 (29.5)
6 months–1 year	14 (26.9)	4 (7.5)	18 (17.1)
1–5 years	14 (26.9)	23 (43.4)	37 (35.2)
5–10 years	1 (1.9)	5 (9.4)	6 (5.7)
>10 years	1 (1.9)	7 (13.2)	8 (7.6)
No answer	2 (3.8)	0 (0)	2 (1.9)
Other chronic conditions	38 (73.1)	45 (84.9)	83 (79)

Columns are expressed as: count (%). Parameters list diagnosis leading to PT. Counts are provided for reasons participants are provided PT without a hypogonadism diagnosis. Other health and clinical characteristics are listed. Percentages in rows have been rounded to the nearest 10th.

**Table 4 ijerph-16-03261-t004:** Frequency of coding themes related to why men sought PT.

Codes, (%)	Respondents <40 Years (*n* = 52)	Respondents ≥40 Years (*n* = 53)	Totals (*n* = 105)
Low testosterone	17 (32.7)	22 (41.5)	39 (37.1)
Well being	20 (38.5)	17 (32.1)	37 (35.2)
Energy	13 (25)	17 (32.1)	30 (28.6)
Libido ^a^	4 (7.7)	19 (35.8)	23 (21.9)
Social energy	10 (19.2)	10 (18.9)	20 (19.0)
Fat	8 (15.4)	11 (20.8)	19 (18.1)
Doctor	8 (15.4)	9 (17)	17 (16.2)
Other/Misc.	10 (19.2)	4 (7.5)	14 (13.3)
Depression	3 (5.8)	8 (15.1)	11 (10.5)
Mood	4 (7.7)	2 (3.8)	6 (5.7)
Muscle	2 (3.8)	4 (7.5)	6 (5.7)
Classical hypogonadism	1 (1.9)	4 (7.5)	5 (4.8)
Erectile dysfunction	1 (1.9)	4 (7.5)	5 (4.8)
Focus	2 (3.8)	2 (3.7)	4 (3.8)
Relationship	1 (1.9)	3 (5.7)	4 (3.8)
Sex (other)	1 (1.9)	3 (5.7)	4 (3.8)
Youth	1 (1.9)	2 (3.8)	3 (2.9)
Anxiety	2 (3.8)	0 (0)	2 (1.9)
Dominance/Assertiveness	2 (3.8)	0 (0)	2 (1.9)
Comorbidities	1 (1.9)	1 (1.9)	2 (1.9)
Cancer	1 (1.9)	0 (0)	1 (<1)
Cognition	1 (4)	0 (0)	1 (<1)
Mate acquisition	1 (1.9)	0 (0)	1 (<1)

Columns are expressed as: count (%). Parameters list codes generated when respondents were asked “what did you perceive as the benefits of testosterone”. ^a^ Differences in counts between older and younger men (*p* < 0.01, calculated using Fisher’s exact test, α = 0.05). Codes’ definitions are found in the codebook, in Table 1. Percentages in rows have been rounded to the nearest 10th.

**Table 5 ijerph-16-03261-t005:** Frequency of coding themes related to beneficial effects noticed after PT.

Codes, (%)	Respondents <40 Years (*n* = 52)	Respondents ≥40 Years (*n* = 53)	Totals (*n* = 105)
Energy	24 (46.2)	31 (58.5)	55 (52.4)
Libido	19 (36.5)	25 (47.2))	44 (41.9)
Muscle	13 (25)	17 (32.1)	30 (28.6)
Other/Misc.	11 (21.2)	16 (30.2)	27 (25.7)
Fat	10 (19.2)	9 (17)	19 (18.1)
Well being	8 (15.4)	11 (20.8)	19 (18.1)
Social energy	8 (15.4)	10 (18.9)	18 (17.1)
Mood	10 (19.2)	5 (9.4)	15 (14.3)
Sex (other)	5 (9.6)	8 (15.1)	13 (12.4)
Focus	4 (7.7)	8 (15,1)	12 (11.4)
Cognition	6 (11.5)	3 (5.7)	9 (8.6)
Youth	4 (7.7)	5 (9.4)	9 (8.6)
Depression	2 (3.8)	6 (11.3)	8 (7.6)
Erectile dysfunction	3 (5.8)	4 (7.5)	7 (6.7)
Dominance	2 (3.8)	2 (3.8)	4 (3.8)
Low testosterone	2 (3.8)	2 (3.8)	4 (3.8)
Anxiety	2 (3.8)	1 (1.9)	3 (2.9)
Comorbidities	2 (3.8)	0 (0)	2 (1.9)

Columns are expressed as: count (%). Parameters list codes generated when respondents were asked “what did you perceive as the benefits of testosterone?”. All *p*-values were >0.10. Codes’ definitions are found in the codebook, in Table 1. Percentages in rows have been rounded to the nearest 10th.

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
