# Peer review of "Sex, Energy, Well-Being and Low Testosterone: An Exploratory Survey of U.S. Men’s Experiences on Prescription Testosterone"

_ijerph, 2019, doi:10.3390/ijerph16183261_

Round 1

Reviewer 1 Report

The manuscript covers an original point of view and it could be of interest for the IJERPH readers.
However, in my opinion, the article needs minor revisions in order to increase the quality, as some points are not clear or may be explained better.

Some data reported in the results (rows 191-192) regarding Table 2 are not the same of what reported in the Table. Please check them carefully. Household Income percentages in Table 2 both for Respondents < 40 Years and for > 40 Years are not correct (the total percentage is 99.9% and not 100). Also the total percentage is not correct (100.1% and not 100) Please correct them. Moreover, in the second row of the second column a bracket is missing. Please include it.

Ethnicity percentages in Table 2 for Totals are not correct (the total percentage is 100.1% and not 100). Please correct them.

Highest level education percentages in Table 2 for Respondents  > 40 Years is not correct (the total percentage is 99.8 % and not 100). Please correct them. In the total, in the first row, the sign "≤" should be deleted. If 1 is not correct, please insert the exact result.

From the row "student" the Table is difficult to understand for the reader. Please divide each different category and add categories like "other" or "no" in order to arrive to the 100% of total for each category that you analyse.

Data presented at rows 202-204 and some of the data presented at rows 233-234 are not the same presented in Table 3 and 4. Also at row 208 the percentage of older men presenting multiple other chronic conditions is different from the one presented in Table 3. Please uniform them.

Hypogonadism diagnosis percentages and Lenght of time in PT in Table 3 for Respondents < 40 Years, for > 40 Years and totals are not correct (the total percentage is different from 100). PT providers and PT formula percentages in Table 3 for Respondents < 40 Years are not correct (the total percentage is different from 100).

"Other" number count and percentage are really difficult to understand. You write at row 218 that percentages are based on 56 participants but the percentages are not correct. If you gave participants the possibility to select more than one options you should cite and the percentages should be based on this: the first column on the participants under 40, the second column on the participants above 40 years and the third on the total.

At row 243 the percentage indicated for the younger group is different from the one reported in the Table. If the correct one is the one reported in the Table, also at row 242 "almost four times" should be changed in "more than four times". Also some of the percentages at rows 246-247 are different from the ones reported in Table 4. Please check and uniform them. 

Some of the percentages at rows 263-264 are different from the ones reported in Table 5. Please check and uniform them.

I also suggest to include the limitation that sexual orientation of the men were not taken into account, since it is an important variable to consider when talking about motivation to sexuality (you can for example cite this new article on this topic):

Eleuteri, S., Rossi, R., Simonelli, C., 2019, How to address clinical work with older bisexual clients and their partners? Sexologies, in press, doi: 10.1016/j.sexol.2019.05.010.

Author Response

The manuscript covers an original point of view and it could be of interest for the IJERPH readers.
However, in my opinion, the article needs minor revisions in order to increase the quality, as some points are not clear or may be explained better. 

Thank you very much for the constructive feedback and interest for our manuscript. Minor changes have been made throughout the manuscript to ease readers and correct typographical discrepancies. Please see tracked changes for such changes.

Some data reported in the results (rows 191-192) regarding Table 2 are not the same of what reported in the Table. Please check them carefully. Household Income percentages in Table 2 both for Respondents < 40 Years and for > 40 Years are not correct (the total percentage is 99.9% and not 100). Also the total percentage is not correct (100.1% and not 100) Please correct them. Moreover, in the second row of the second column a bracket is missing. Please include it.

Ethnicity percentages in Table 2 for Totals are not correct (the total percentage is 100.1% and not 100). Please correct them.

Thank you for pointing out errors in the calculation of some percentages in our manuscript. We have acknowledged that some of our percentages may not add to 100% and this looks to be a result of rounding, due to our sample size of 105. That said, your response highlights that rounding may be overlooked by a reader and thus we have inserted a footnote at the bottom of each table which contains descriptive statistics: “Percentages in rows have been rounded to the nearest 10th.”. We hope this assists when reading our tables.

Data presented at rows 202-204 and some of the data presented at rows 233-234 are not the same presented in Table 3 and 4. Also at row 208 the percentage of older men presenting multiple other chronic conditions is different from the one presented in Table 3. Please uniform them.

Hypogonadism diagnosis percentages and Lenght of time in PT in Table 3 for Respondents < 40 Years, for > 40 Years and totals are not correct (the total percentage is different from 100). PT providers and PT formula percentages in Table 3 for Respondents < 40 Years are not correct (the total percentage is different from 100).

"Other" number count and percentage are really difficult to understand. You write at row 218 that percentages are based on 56 participants but the percentages are not correct. If you gave participants the possibility to select more than one options you should cite and the percentages should be based on this: the first column on the participants under 40, the second column on the participants above 40 years and the third on the total.

Thank you for pointing out errors in the calculation of some percentages in our manuscript, particularly when transferring them from a table to the text. Table 3 in particular seemed to have a major discrepancy with the “other category”, which may cause confusion for readers. Since this “other” section (multiple responses were in fact given, but upon recalculating them in this one category it just seemed confusing to read) it seemed unnecessary to include a full section in the table to describe this supplemental information.  Additionally, a line in section 3.2 was added to make it clearer what it refers to: “. Patients described, in open-ended responses that they received PT from a physician for the following reasons, excluding a hypogonadism diagnosis : “low T” (48.6%), energy (10.5%), and libido (17.8%)."Further cuts were made to section 3.2 to remove data analyses which did not seemed like it would be distracting to readers. All rounding was also recalculated in each section of the analyses and typographical errors, particularly in percentages, were corrected (and rounded to the nearest 10th).

At row 243 the percentage indicated for the younger group is different from the one reported in the Table. If the correct one is the one reported in the Table, also at row 242 "almost four times" should be changed in "more than four times". Also some of the percentages at rows 246-247 are different from the ones reported in Table 4. Please check and uniform them.

Some of the percentages at rows 263-264 are different from the ones reported in Table 5. Please check and uniform them.

Thank you for pointing out errors in table 3, which prompted a more careful analyses of percentages throughout the paper. All should now be uniform and more accurately reflect counts contained in the tables and dataset. Your feedback was very valuable to adding a required extra polish to sections of the manuscript, which did not stick out prior.

I also suggest to include the limitation that sexual orientation of the men were not taken into account, since it is an important variable to consider when talking about motivation to sexuality (you can for example cite this new article on this topic):

Eleuteri, S., Rossi, R., Simonelli, C., 2019, How to address clinical work with older bisexual clients and their partners? Sexologies, in press, doi: 10.1016/j.sexol.2019.05.010.

Sincerely, thank you for reminding us of the limitation in omitting questions involving sexual orientation. We acknowledge that this is a major limitation of the study and was a point of conflict during the development of our survey. Due to the hypothetical novelty PT treatment introduces to male aging and social context, we originally thought that adding even further nuance to our questionnaire (in this case sexuality but we also considered gender identity) might overwhelm the analysis. That said, in retrospect we regret leaving out sexuality, as this is a crucial and changing factor in men. The paper you provided also helped remind us in the realities of how sexuality might affect the well-being of men seeking treatments such as testosterone therapy, especially cross-culturally, where there can be significant variation in how patients experiments every stage of a medical treatment. A citation has and about 2 lines have been added to the limitation section of our discussion. Additionally, the citation you shared is once again cited in the conclusion, as it points out the critical role of sexuality in social context. Thank you so much for pointing this out and we can ensure this factor will be recorded in future efforts going forward.

Reviewer 2 Report

Sex, Energy, Well-Being and Low Testosterone: An Exploratory Survey of U.S. Men’s Experiences on Prescription Testosterone

General

This is a well written, agreeable to read text. The subject is interesting, although not interesting a large public.

As the authors state, this survey is the first attempt to ask PT patients to report their experiences without being prompted by closed-ended questions.

The methodology is strong, and limitations are correctly reported in the discussion.

My major concern is the definition of ‘low T’. How can participants feel they have a low testosterone level? Because no real values are asked for. Nor proof.

The discussion is lengthy, and quite philosophical and hypothetical at times. I would suggest to shorten it, and to stick to facts.

Please limit the number of abbreviations to three. The text becomes quite difficult to read due to all those acronyms.

Details

31

$70 million in 2000 to more than $2 billion in 2013

Contrast with:

38 with sales having risen almost threefold between 2000 and 2013

123

Recruitment and data collection began after the study was deemed ‘except’ by the University 124 of Nevada

Deemed ‘accepted’??

125

‘Prior’ is written twice.

168 Table 1

‘without a participant’s belief of the ‘present’ of Low T.’ Should be ‘presence’?

189

Strange that the oldest man was only 66 years old. Why do we not see participants up to 85?

386

PT may be allowing men the energy to maintain engagement in their domestic and 387 professional responsibilities, something which may be missed by lab studies.

<this is highly hypothetical, and should be proven by random double-blind trials.

404

marred),

should be ‘married’

Conclusion

‘We suspect there is a subtler 472 interaction between energy/social energy than can be typically observed in clinical settings and 473 highlight the need to consider the effects of partner, childcare, and coalitional dynamics on male’s 474 perceptions of low T and the effects of treatment.’

<I propose to stick to real results, and to not include some hypotheses in the conclusion.

Author Response

This is a well written, agreeable to read text. The subject is interesting, although not interesting a large public.

As the authors state, this survey is the first attempt to ask PT patients to report their experiences without being prompted by closed-ended questions.

The methodology is strong, and limitations are correctly reported in the discussion.

Thank you very much for the positive feedback on the writing and methodology, as well as pointing out the niche nature of the research question. That said, we believe it is a worthwhile investigation, and may bring some clarity or direction to the investigations concerning testosterone therapy and how patients experience the treatment. Particularly those undergoing off-label prescription testosterone treatment, in the United States.

My major concern is the definition of ‘low T’. How can participants feel they have a low testosterone level? Because no real values are asked for. Nor proof.

Thank you for the feedback, this was also a concern of ours and one we discussed. Many factors in logistics themselves, such as having an accurate retrospective account of one or multiple blood values, taken at multiple times may have led to incomplete data and an even smaller sample. But largely, it was the controversies and between individual and population variation on what constitutes as androgen deficiency which us to take more interest in a subjective account of low testosterone. More detail below:

We decided to opt for a subjective account of “low T” due to the varying diagnostic criteria and disagreement over what constitutes as a low testosterone blood value. On page 2 44-49 literature here discusses controversies surrounding identifying a low T threshold and the variation in prescribing patterns related to this. It may be a series of factors related to the ambiguous nature of “low T” as a diagnostic criterion. There is evidence in the cited literature that prescribing can vary in nature from full compliance with a Urology or Endocrine Guidelines, to prescribing without any lab tests. Additionally, the symptoms that present at “low T” values (See: Bhasin et al., 2018 for a review) may vary between individuals and this seems to hold true for the effects “low t” patients notice after treatment, even if they were below reference ranges. Due to the drastic (both clinical and between individual/population) variation on what constitutes low testosterone, we chose to leave a question for reference ranges out and instead focused on subjective experiences of testosterone patients. Lines around 298-309 try to address this. That all said, many themes such related to sex and energy seem salient with literature on Androgen deficiency.

The discussion is lengthy, and quite philosophical and hypothetical at times. I would suggest to shorten it, and to stick to facts.

The discussion was shortened at several places, where it seemed excessively speculative. Further details are pointed out below.

Please limit the number of abbreviations to three. The text becomes quite difficult to read due to all those acronyms.

Acronyms used are now PT, LOH, and ED. U.S. is also used in place of United States in parts, under the assumption that the convention is familiar to readers.

Details

Thank you so much for these below details. The changes discussed below are found in tracked changes. The paper was additionally altered in places to clarify topics to the reader and remove formatting errors.

31

$70 million in 2000 to more than $2 billion in 2013

Contrast with:

38 with sales having risen almost threefold between 2000 and 2013

Thank you for pointing out how this could be a bit confusing to read. Here we were trying to highlight the growth of the United States testosterone sales compared to the rest of the globe. In order to stay concise, the sentence was shorted to get more to the point.

123

Recruitment and data collection began after the study was deemed ‘except’ by the University 124 of Nevada

Deemed ‘accepted’??

Thank you for finding that typographical error. At our University “Exempt” is used by our IRB in place of accepted. This change has now been made.

125

‘Prior’ is written twice.

Prior is now only written once.

168 Table 1

‘without a participant’s belief of the ‘present’ of Low T.’ Should be ‘presence’?

This has been corrected, thank you.

189

Strange that the oldest man was only 66 years old. Why do we not see participants up to 85?

We suspect this was an admittedly frustrating aspect of convenience sampling. Since this survey was accessed through an online link, that there may be a bias skewing the participants to the younger side. Additionally, it may be that 66+ year old’ are more difficult to find. Due to a hesitance in prescribing as men age and may have more complications [see: 30 for reviews on age related issues]. Additionally, it maybe that middle-aged men might be overrepresented due to the growth in prescribing in that age range [33]. The discussion and background briefly touch on this but nothing further was added now that we tried to remove more speculation.

386

PT may be allowing men the energy to maintain engagement in their domestic and 387 professional responsibilities, something which may be missed by lab studies.

<this is highly hypothetical, and should be proven by random double-blind trials

This line has been removed. While domestic and professional responsibilities may have a role in the context PT is experienced, this study upon review can not add to that conversation with its current limitations. To further shorten the discussion, this section was cut. In addition to some parts around 365-373 and 471-473. Tracked changes should reflect this.

404

marred),

should be ‘married’

Corrected

Conclusion

‘We suspect there is a subtler 472 interaction between energy/social energy than can be typically observed in clinical settings and 473 highlight the need to consider the effects of partner, childcare, and coalitional dynamics on male’s 474 perceptions of low T and the effects of treatment.’

<I propose to stick to real results, and to not include some hypotheses in the conclusion.

Some adjustments have been made in the conclusion to minimize speculation, while also still broadly highlighting aging and social context as further areas of interest when studying PT.